# PAIRNet: Predicting PIWI cleavage specificity via position-aware RNA interaction modeling

**Lin Zeng**[1], **Zhenzhen Li**[2], **Enzhi Shen**[2]\*, **Shikui Tu**[1]\*, **Lei Xu**[1]

**1** Center for Cognitive Machines and Computational Health (CMaCH), School of Computer Science, Shanghai Jiao Tong University, Shanghai, China, **2** Key Laboratory of Growth Regulation and Translational Research of Zhejiang Province, School of Life Sciences, Westlake University, Hangzhou, Zhejiang, China

\* tushikui@sjtu.edu.cn (ST); shenenzhi@westlake.edu.cn (ES)

## Abstract

PIWI proteins maintain genome integrity by piRNA-guided cleavage of complementary RNA targets. While Cleave-N'-Seq (CNS-seq) has advanced our understanding of PIWI targeting logic through quantitative mapping of cleavage rates and pairing rules, its labor-intensive workflows hinder systematic exploration of sequence determinants. Here, we present PAIRNet, a deep learning framework that predicts PIWI-mediated RNA cleavage rates by explicitly modeling guide-target interactions. Recognizing that interaction geometry, not just sequence, dictates cleavage efficiency, PAIRNet integrates biochemical insights with computational innovation: it encodes pairing states, mismatch types, insertions, and deletions alongside learnable positional embeddings to quantify spatial dependencies; employs a hybrid CNN-Transformer architecture prioritizing duplex dynamics over static sequence features to resolve both local catalytic motifs (e.g., contiguous base-pairing at g10–g11) and distal structural perturbations; and incorporates interpretability modules (saliency maps, counterfactual analysis) to link interaction patterns to biochemical insights and uncover position-specific cleavage rules. Validated across four PIWI-guide datasets, PAIRNet consistently ranks among the top two performers in all experimental conditions, achieving the most pronounced relative improvements in PCC, 34.7% for MILI and 14.6% for MIWI, over second-ranking methods. Critically, PAIRNet recapitulates key biological principles—stringent complementarity at catalytic residues (g10–g11) and tolerance for 3' mismatches—aligning with structural studies of PIWI dynamics. By bridging biochemical precision with computational scalability, PAIRNet establishes a roadmap for designing high-specificity piRNA silencing tools while accelerating mechanistic studies of RNA-guided genome defense.

**Data availability statement:** The source code and data used to produce the results and analyses presented in this manuscript are available from https://github.com/CMACH508/PAIRNet.

**Funding:** This work was supported by the National Natural Science Foundation of China (Grant No. 62172273 to ST), the Science and Technology Commission of Shanghai Municipality (Grant No. 24510714300 to ST), and the Shanghai Municipal Science and Technology Major Project, China (Grant No. 2021SHZDZX0102 to ST). The funders had no role in study design, data collection and analysis, decision to publish, or preparation of the manuscript.

**Competing interests:** The authors have declared that no competing interests exist.

## Author summary

Small RNA systems like miRNA and siRNA—honored by Nobel Prizes—play vital roles in biology. As another key member, piRNA works with PIWI proteins to defend genomes by cutting viruses and mobile genetic elements. However, unlike miRNA and siRNA, piRNA's targeting rules are far more complex, limiting its therapeutic potential. While recent studies use labor-intensive experiments like Cleave-N'-Seq to decode these rules, reproducing such work requires simulating biochemical reactions across multiple time points—a slow and technically challenging process. To solve this, we developed a computational method that focuses on how piRNA and target RNAs physically interact, rather than just analyzing sequences. Traditional approaches, which stitch sequences together or count generic patterns (e.g., *K*-mers), achieved low accuracy. In contrast, our interaction-centric model—designed to mirror natural PIWI behavior—not only improved predictions by up to 34.7% over conventional tools, but also identified critical rules, such as strict pairing at catalytic sites and flexibility elsewhere. This success demonstrates that modeling biological systems should start with the problem's essence, not just sequences. By aligning computation with nature's logic, we can accelerate piRNA-based therapies and inspire smarter tools for genome engineering.

## Introduction

Small noncoding RNAs, such as microRNAs (miRNAs), small interfering RNAs (siRNAs), and piwi-interacting RNAs (piRNAs), have transformed our understanding of gene regulation by acting as guides within RNA-induced silencing systems. While miRNAs and siRNAs have garnered extensive study—yielding Nobel Prize-winning breakthroughs in 2006 and 2024—piRNAs are now stepping into the spotlight as a vital class of small RNAs. Spanning 24 to 32 nucleotides, piRNAs primarily silence transposable elements (TEs) to shield the genome from transposon-driven instability [1,2]. In mammals, piRNAs partner with PIWI proteins to form piRNA-induced silencing complexes (piRISCs), which orchestrate the precise recognition and cleavage of complementary RNA targets, such as TE transcripts, safeguarding germline genome stability [3–7]. Beyond this, piRNAs also target viral RNA genomes, bolstering antiviral defenses [8–10]. These roles underscore piRNAs' critical contributions to genetic and cellular integrity, particularly in reproduction and immunity, driven by their guided recognition and cleavage capabilities.

While miRNA, siRNA, and piRNA are all small RNAs involved in gene regulation, their working mechanisms diverge significantly, with piRNA-guided silencing presenting a more intricate and less predictable paradigm compared to the relatively well-defined rules governing miRNAs and siRNAs. miRNAs, typically 22 nucleotides long, guide AGO-clade Argonaute proteins by relying on canonical 5' seed pairing (nucleotides g2–g8, where **g** denotes guide RNA positions numbered from the 5' end), which often suffices for target repression [11]. siRNAs, usually 21 nucleotides long, enforce strict complementarity across their entire length to direct AGO-mediated

cleavage [12,13]. Both miRNA and siRNA hinge on precise molecular rules, seed dominance for miRNAs, perfect pairing for siRNAs, making their targeting logic largely interpretable. In contrast, piRNAs guide PIWI-clade Argonaute proteins through a dynamic and context-dependent mechanism [14]. PIWI proteins exhibit remarkable flexibility: they bind targets independent of canonical seed pairing and tolerate mismatches even within extended complementary regions [14]. While PIWI-catalyzed slicing requires at least 15 contiguous base pairs, longer guide-target duplexes paradoxically allow mismatches at virtually any position, defying the rigid rules seen in other RNA silencing [14]. This mechanistic plasticity enables PIWI-piRNA complexes to efficiently cleave partially paired transcripts—a critical adaptation for silencing rapidly evolving transposons, whose sequences often evade detection by more stringent systems [14]. Yet, this very flexibility renders piRNA targeting logic less straightforward, demanding nuanced biochemical and computational approaches to unravel its complexity.

Cleave-'n-Seq (CNS-seq) [14,15], with its high-throughput capabilities, plays a crucial role in this study and other related works [16–19], offering in-depth insights and strong evidence for several key findings. The three-step CNS-seq (Fig 1A) first constructs a library of variants of RNA targets (pairing, mismatch, insertions, deletions, etc.) and assembly piRISCs (piRNA-induced silencing complexes); subsequently, it incubates the guide piRISCs with the constructed RNA target library to initiate cleavage reactions for multiple defined time intervals, and high-throughput techniques are used to detect the cleavage products and quantifies the cleaved products; finally, the pre-steady-state cleavage rates ($k$) of thousands of target variants are determined by fitting the products abundance to burst-and-steady-state scheme formula. By simultaneously analyzing the cleavage rates of thousands of target variants, CNS-seq enables the researchers to comprehensively assess how PIWI proteins, such as MILI, MIWI, respond to different degrees and types of mismatches. This high-throughput approach allows for the identification of patterns in cleavage efficiency across a wide range of target RNAs.

CNS-seq, despite its value in studying PIWI-mediated RNA cleavage, has several limitations. The experimental procedures are highly complex, requiring intricate RNA library construction and precise molecular techniques, making validation challenging. The process is also time-consuming and labor-intensive, involving meticulous monitoring of cleavage reactions, extensive data analysis, and significant hands-on effort from researchers. More importantly, studying multiple PIWI proteins necessitates testing numerous guide-target combinations, exponentially increasing the complexity, resource demands, and limiting scalability for broader applications. For those proteins that are difficult to purify, it is impossible to conduct a complete CNS-seq experimental analysis. When revisiting MILI/MIWI data in Gainetdinov et al. [14], although we successfully reproduced the change in pre-steady-state cleavage rate for single (S1A Fig) or double consecutive mismatches (S1B Fig) for contiguous g2-g21 positions, we still observed variations between different guide RNA both for MILI (S1C Fig) and MIWI (S1D Fig). Specially, We observed a clear deviation in cleavage performance when loading Kctd7 or piRNA2 compared to L1MC and piRNA1 at the g10 (S1D Fig). These guide-specific disparities—unexplained by prior analyses—suggest that nucleotide identity and positional context jointly influence cleavage efficiency, a hypothesis intractable to test experimentally with only four unique guide RNAs. This limitation motivates the AI-driven approach of PAIRNet: By modeling nucleotide-level interactions at all guide positions, the framework generalizes beyond sparse experimental data to infer how sequence composition and spatial pairing jointly dictate the specificity of PIWI.

To address the limitations of CNS-seq while retaining its quantitative power, we trained PAIRNet on *in vitro* cleavage data [14] to predict PIWI-mediated RNA cleavage rates based on guide-target interaction patterns (Fig 1B). PAIRNet's framework bridges biochemical precision with computational scalability through three key innovations.

First, it replaces naïve sequence-encoding paradigms, such as concatenated one-hot vectors, *K*-mer frequencies, or CNN-based [20] embeddings (Fig 1C), with interaction-centric feature engineering. These traditional sequence-based encoding methods often treat the sequences independently, struggling to model the critical positional interactions between guide and target RNAs that drive cleavage efficiency. To overcome this, PAIRNet explicitly encodes positional pairing states, mismatch types, insertions, and deletions (Fig 1D). This interaction-centered encoding demonstrated a clear

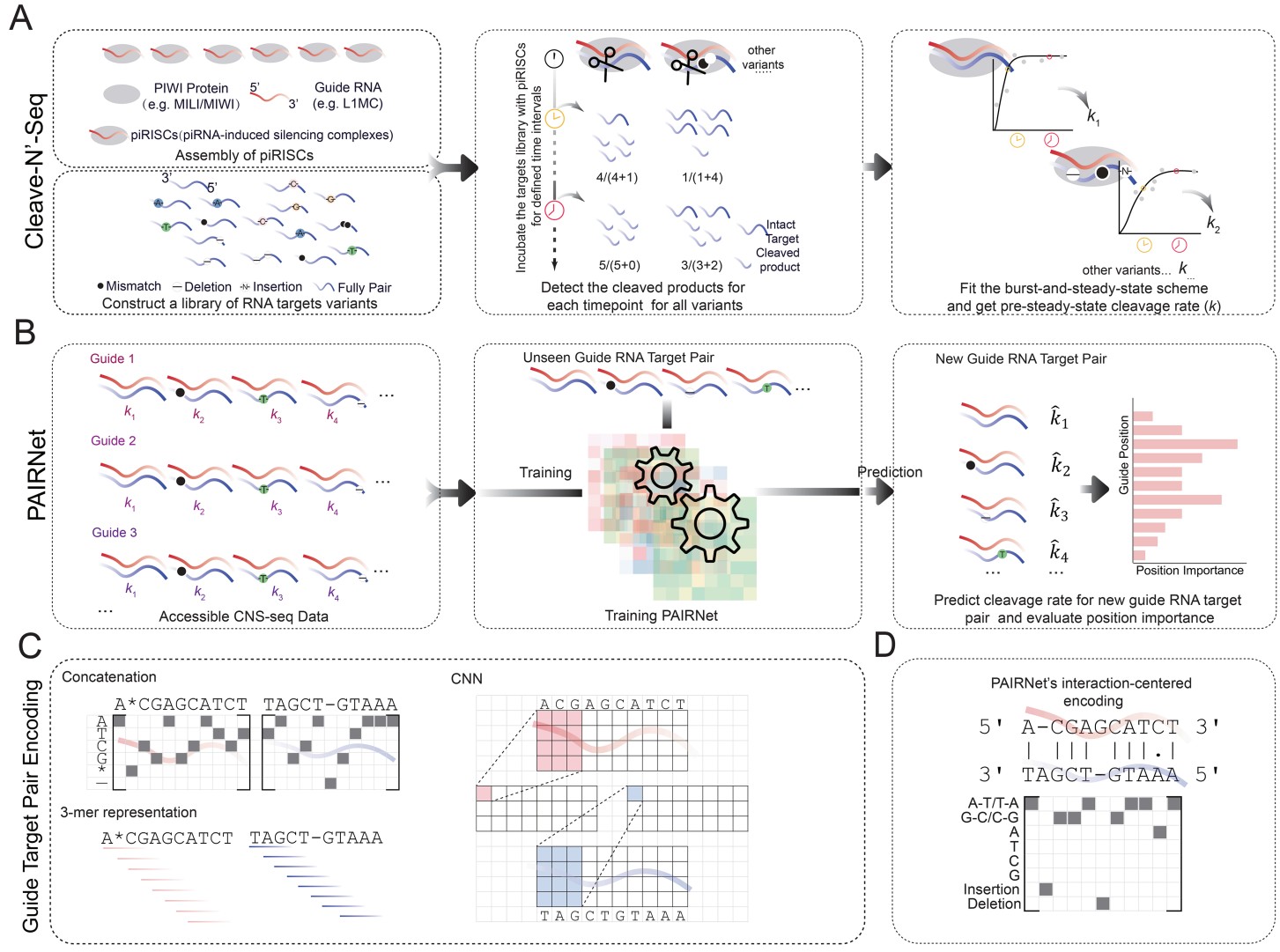

**Fig 1. Cleave-N'-Seq (CNS-Seq) workflow and PAIRNet prediction model.** (A) CNS-Seq experiment workflow. The CNS-Seq protocol involves assembly of piRISCs in vitro and preparation of RNA targets variants pool, then incubation of guide-target complexes to induce RNA cleavage, and cleavage rate ($k$) calculation based on inferred products abundance. (B) PAIRNet predicts PIWI cleavage specificity. PAIRNet is one machine learning model leverages accessible CNS-Seq-derived $k$ values and guide target sequence pair for predicting unseen new guide-target cleavage rate $\hat{k}$, enabling high-throughput computational screening of functional pairs and reducing reliance on labor-intensive experimental validation, and position importance analysis provides biological insight of rules governing cleavage. (C) Sequence-based encoding strategies (concatenation of one-hot encoding, 3-mer representation and CNN-based extraction) and PAIRNet's interaction-centered encoding. (D) Interaction-centered encoding.

advantage over sequence-based methods across MIWI and MILI datasets, consistently achieving best prediction performance regardless of downstream model. Additionally, each guide position is mapped to a learnable embedding vector that captures its spatial hierarchy, enabling the model to distinguish critical positions from non-essential regions.

Second, PAIRNet leverages a hybrid CNN-Transformer architecture to jointly model local duplex stability and global structural dependencies. The 1D convolutional layers detect catalytic-site motifs (e.g., contiguous Watson-Crick pairs at g10–g11), while transformer [21] layers resolve distal interactions influenced by insertions or deletions. This design mirrors PIWI's structural dynamics, where local pairing stabilizes the catalytic core and global flexibility accommodates

mismatches [14,22]. The hybrid architecture, together with interaction-centered encodings, consistently ranks among the top two performers across all datasets, achieving the most pronounced improvements of 34.7% for MILI (dataset3) and 14.6% for MIWI (dataset2) in Pearson Correlation, while maintaining robust accuracy in all experimental contexts, reflecting its capacity to decode PIWI's dynamic RNA targeting logic.

Third, PAIRNet quantifies position-specific cleavage rules through interpretable saliency maps and counterfactual analysis. Learnable positional embeddings and gradient-based scoring identify critical positions (e.g., g10–g11). The model recapitulates known biological priors, such as stringent pairing requirements at the catalytic core and tolerance for 3' mismatches [23], offering actionable insights for piRNA guide design.

Together, these features allow PAIRNet to address the limitations of sequence-only models, demonstrating improved predictive accuracy across diverse datasets while offering interpretable insights into PIWI targeting preferences. By connecting computational predictions to structural principle, such as catalytic core rigidity and 3' flexibility, PAIRNet supports the design of specific piRNA guides and contributes to our understanding of PIWI targeting mechanisms. This study demonstrates the value of modeling interaction geometry in RNA-guided silencing, offering a computational approach that could inform the development of future RNA-targeting tools. To facilitate community use, we have implemented PAIRNet as a user-friendly web server available at https://scgerm-atlas.sjtu.edu.cn/pairnet/pairnet.html, which automates the feature encoding and cleavage rate prediction for custom guide-target pairs.

## Materials and methods

### Problem setting

Let $\mathcal{D} = \{(g_i, t_i, k_i)\}_{i=1}^N$ denote a dataset of $N$ RNA guide-target pairs, where $g_i \in \{A, C, G, U\}^{26}$ is the guide RNA sequence, $t_i \in \{A, C, G, U\}^m$ is the target sequence, and $k_i \in \mathbb{R}^+$ is the pre-steady-state cleavage rate, a measure of cleavage efficiency calculated by fitting time-course cleavage product ratios to a burst-and-steady-state kinetic model [14]. In simpler terms, we aim to predict how efficient a PIWI protein cuts a target RNA based on its match with a guide RNA *in vitro* (Fig 1B). This involves exploring the pairing rules and quantifying how mismatches at different positions affect cleavage efficiency, using data from CNS-seq experiments. We formulate a regression task to learn a function $f(g, t) \rightarrow \hat{k}$ that predicts cleavage rates while explicitly modeling position-dependent pairing interactions.

### Overview of method

An overview of the proposed method is depicted in Fig 2. The key features of our framework lie in a position-aware encoding on the guide-target RNA pair data and a hybrid CNN-Transformer network for the representation learning on the interaction patterns. Specifically, PAIRNet begins by explicitly encoding guide-target interactions—including Watson-Crick pairing, mismatches, insertions, and deletions—at each guide position, augmented with learnable positional embeddings that prioritize critical regions like the catalytic core (g10–g11) while deprioritizing non-essential sites (Fig 2A). This spatial hierarchy is then processed through a hybrid CNN-Transformer architecture: convolutional layers detect local motifs (e.g., contiguous base-pairing near catalytic residues, Fig 2B), while transformer layers resolve global dependencies (e.g., structural disruptions caused by distal insertions or deletions, Fig 2C), mirroring PIWI's dual requirements for local precision and global adaptability. Finally, the framework applies multi-layer network to predict cleavage rate $\hat{k}$, and concludes with interpretable outputs—saliency maps highlighting position-specific contributions to cleavage efficiency and counterfactual analysis quantifying mismatch tolerance—providing biochemical insights into PIWI targeting rules while achieving high predictive accuracy (Fig 2D).

### Interaction-centric encoding

Traditional sequence-based encoding methods (e.g., via concatenated one-hot vectors, CNN-based or *K*-mer frequencies) often treat RNA as independent sequences, and overlook the dynamic interplay between guide and target RNAs,

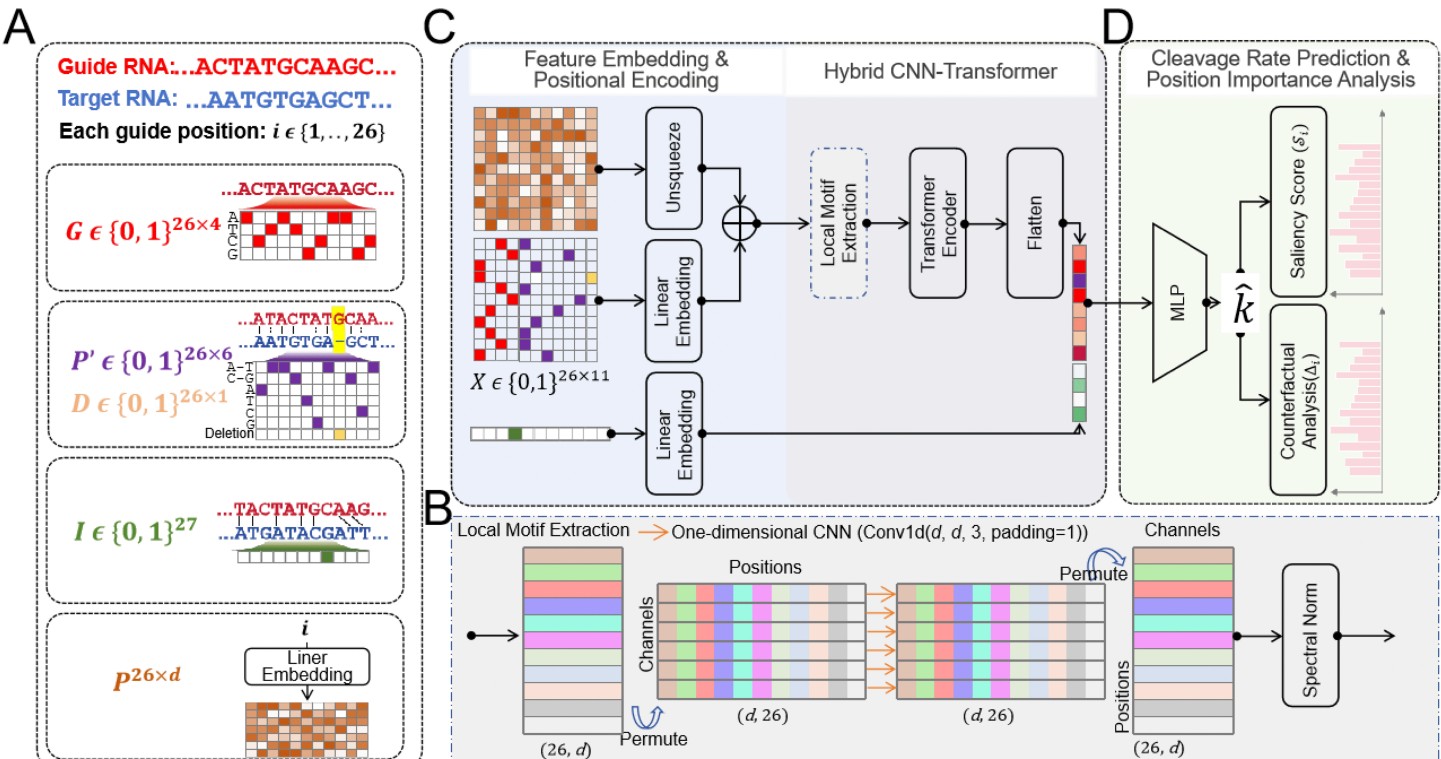

**Fig 2**. Flowchart of PAIRNet. (A) Position-aware interaction-centered encoding. Guide RNA ($\mathbf{G} \in \{0, 1\}^{26\times4}$) and target RNA are encoded into matrices ($\mathbf{P'}$, $\mathbf{D}$, $\mathbf{I}$) representing pairing states, deletions, and insertions. Learnable positional embeddings prioritize catalytic regions (e.g., g10–g11). (B) Local motif extraction details. A 1D convolutional network (Conv1d($d$, $d$, 3)) identifies short-range motifs (e.g., contiguous Watson-Crick pairs and adjacent biological domain). (C) Hybrid CNN and Transformer network for local motif extraction and global dependency modeling. Transformer layers contextualize interactions across the duplex, resolving long-range structural effects (e.g., distal insertions). (D) Prediction and interpretability. A multi-layer network predicts cleavage rates ($\hat{k}$), while saliency scores ($S_i$) and counterfactual analysis ($\Delta_i$) quantify positional importance and mismatch impacts.

missing critical biochemical cues like mismatch tolerance or structural disruptions that define PIWI-mediated cleavage. Motivated by the need to transcend simplistic sequence-based paradigms, PAIRNet adopts an interaction-centric encoding strategy that directly quantifies guide-target pairing dynamics—a critical advance over conventional approaches (Fig 2A). For example, mismatches are categorized by nucleotide identity, enabling the model to distinguish biochemically distinct scenarios (e.g., a G-U wobble versus a non-canonical A-A clash). This granular encoding mirrors the flexibility of PIWI proteins, which tolerate certain mismatches while enforcing strict complementarity at catalytic residues. By prioritizing interaction geometry over raw sequence, PAIRNet bridges a key gap between computational abstraction and biological reality, laying the foundation for accurate cleavage rate prediction.

**RNA sequence representation.** To disentangle the roles of guide sequence and interaction patterns in cleavage outcomes, PAIRNet represents the guide and target RNAs separately, allowing the model to weigh their independent contributions—a key step toward accurate prediction.

Guide RNA: Represented as a one-hot matrix $\mathbf{G} \in \{0, 1\}^{26\times4}$, where each row corresponds to one of the 26 positions in the guide RNA sequence, and each column represents one of the four possible nucleotides (A, U, C, G), for each position $i \in \{1, \dots, 26\}$:

$$\mathbf{G}_{i,j} = \begin{cases} 1 & \text{if nucleotide } j \text{ (A, U, C or G) is present at position } i \\ 0 & \text{otherwise} \end{cases}$$

Target RNA: Processed into three feature matrices $\mathbf{P}' \in \{0, 1\}^{26\times3}$ (pairing states), $\mathbf{D} \in \{0, 1\}^{26\times1}$ (deletions), and $\mathbf{I} \in \{0, 1\}^{27}$ (insertions). The pairing matrix $\mathbf{P}'$ is derived from combining match features $\mathbf{m}_i \in \mathbb{R}^2$ and mismatch vectors $\mathbf{e}_i \in \mathbb{R}^4$, as detailed in the next section.

**Guide-target encoding.** To capture the full spectrum of guide-target interactions, PAIRNet encodes four critical features at each position—canonical base pairing, mismatch types, deletions, and insertions—reflecting the complex rules PIWI proteins follow in tolerating imperfections while targeting RNA. For each position $i$ in the guide:

- **Watson-Crick Pairing:** Canonical base-pairing modes (A-U vs. C-G) exhibit distinct thermodynamic stabilities and binding geometries that directly influence duplex formation. For instance, C-G pairs form three hydrogen bonds compared to A-U's two, conferring greater stability. To quantify these differences, we explicitly model Watson-Crick matches as separate states, match features $\mathbf{m}_i \in \mathbb{R}^2$ encode:

$$\mathbf{m}_i = \begin{cases} [1, 0] & \text{A-U pairing} \\ [0, 1] & \text{C-G pairing} \\ [0, 0] & \text{otherwise} \end{cases}$$

- **Mismatch Typing:** Beyond canonical matches, mismatch identity (e.g., A-C vs. G-U) profoundly impacts cleavage efficiency. For example, G-U wobble pairs retain partial stability, while A-A clashes disrupt duplex geometry. To dissect these biochemical nuances, we encode mismatches by nucleotide identity, mismatch vector $\mathbf{e}_i \in \mathbb{R}^4$ encodes non-canonical pairings:

$$\mathbf{e}_i[j] = \begin{cases} 1 & \text{Target has nucleotide } j \text{ (A,C,G,U)} \\ 0 & \text{otherwise} \end{cases}$$

Crucially, we encode the specific identity of the mismatched target nucleotide (A, C, G, or U) rather than utilizing a generic binary mismatch flag. By processing this target identity vector ($P'$) alongside the guide nucleotide identity ($G$) within the neural network, PAIRNet implicitly learns specific base-pair representations (e.g., distinguishing a destabilizing A-A clash from a structurally tolerant G-U wobble).

- **Deletion Handling:** Unpaired target nucleotides (deletions) destabilize the guide-target duplex by introducing structural gaps. To flag these perturbations, we employ a binary deletion indicator, deletion indicator $d_i \in \{0, 1\}$:

$$d_i = \begin{cases} 1 & \text{No paired target nucleotide} \\ 0 & \text{Paired} \end{cases}$$

- **Insertion Detection:** Insertions between guide positions induce torsional stress or bulges, altering global duplex conformation. We explicitly model these structural disruptions via an insertion vector, insertion vector $\mathbf{I} \in \{0, 1\}^{27}$ flags insertions between guide positions:

$$\mathbf{I}_j = \begin{cases} 1 & \text{Insertion between guide positions } j \text{ and } j+1 \\ 0 & \text{Otherwise} \end{cases}$$

This detailed breakdown ensures the model reflects the biochemical reality of PIWI flexibility, a cornerstone for predicting cleavage rates with precision.

**Feature embedding and positional encoding.** To integrate spatial context into the interaction features, PAIRNet embeds them with learnable positional encodings, prioritizing key regions like the catalytic core (g10-g11) over less critical ones—an approach inspired by the positional hierarchy in PIWI targeting.

- **Guide Feature Projection:** The guide and pairing features are combined into a single vector $\mathbf{X}_i = [\mathbf{G}_i, \mathbf{P}_i] \in \mathbb{R}^{11}$ for each position $i$, then projected:

$$\mathbf{h}_i^{(0)} = \mathbf{W}^G \mathbf{X}_i + \mathbf{b}^G, \quad \mathbf{W}^G \in \mathbb{R}^{11 \times d}$$

where $\mathbf{h}_i^{(0)}$ represents the initial feature embedding at position $i$, $\mathbf{W}^G$ is a learnable weight matrix, and $\mathbf{b}^G$ is a learnable bias term that adjusts the baseline activation of the embedding. This linear transformation captures nucleotide identity and pairing states while allowing the model to adaptively scale feature importance.

- **Positional Embedding:** Learnable position-specific vectors $\mathbf{p}_i \in \mathbb{R}^d$ are added:

$$\mathbf{h}_i^{(1)} = \mathbf{h}_i^{(0)} + \mathbf{p}_i, \quad \mathbf{p}_i = \text{Embed}(i)$$

where $\mathbf{h}_i^{(1)}$ denotes the position-augmented embedding, enhancing the model's ability to distinguish critical positions (e.g., catalytic core g10-g11) from non-essential regions through adaptive spatial prioritization.

This allows the model to adaptively focus on positions that drive cleavage efficiency, enhancing both accuracy and interpretability.

## Hybrid CNN-transformer

PAIRNet's hybrid CNN-Transformer architecture is a deliberate response to PIWI's dual nature—requiring local precision at the catalytic site and global flexibility across the RNA duplex—merging the strengths of convolutional motif detection with transformer-driven dependency modeling. 1D convolutional layers first scan the interaction-encoded features to detect local motifs critical for cleavage, such as contiguous Watson-Crick pairs near the catalytic core (Fig 2B). These layers act as molecular sensors, identifying short-range patterns (e.g., a 3-nucleotide motif spanning g9-g11) that stabilize the PIWI-target complex. Concurrently, transformer layers resolve long-range dependencies, capturing how distal insertions or deletions perturb the duplex structure (Fig 2C). For example, a bulge at g15 might induce torsional stress that propagates to the catalytic site, reducing cleavage efficiency—a phenomenon observable only through global attention mechanisms. The transformer's self-attention weights quantify pairwise positional interactions, mirroring the allosteric communication observed in PIWI's conformational dynamics. This design captures how local stability and global adaptability together dictate cleavage success, a synergy absent in simpler architectures.

**Local motif extraction.** PIWI-mediated cleavage relies on local catalytic motifs—such as contiguous Watson-Crick pairs near the catalytic core (g10–g11)—to stabilize target binding and activate slicing. To detect these motifs, PAIRNet employs a 1D convolutional layer that scans interaction-encoded features ($\mathbf{H}^{(1)}$) across adjacent positions:

$$\mathbf{H}^{\text{local}} = \text{ReLU}(\mathbf{W}^{\text{conv}} * \mathbf{H}^{(1)}), \quad \mathbf{W}^{\text{conv}} \in \mathbb{R}^{d \times d \times 3}$$

Here, $\mathbf{W}^{\text{conv}}$ represents a set of $d$ learnable filters with a kernel size of 3, enabling the model to recognize trinucleotide-scale interaction patterns (e.g., three consecutive Watson-Crick pairs). The output $\mathbf{H}^{\text{local}} \in \mathbb{R}^{26 \times d}$ captures position-specific motif activations, where each row corresponds to a guide position enriched with local duplex stability features critical for cleavage.

**Global contextualization.** While local motifs drive catalytic activation, distal structural perturbations can propagate stress to the catalytic core, reducing cleavage efficiency. PAIRNet's transformer layers model these long-range dependencies by computing pairwise attention weights between all guide positions:

$$\mathbf{H}^{\text{global}} = \text{Transformer}(\mathbf{H}^{\text{local}}), \quad \mathbf{H}^{\text{global}} \in \mathbb{R}^{26 \times d}$$

The transformer refines $\mathbf{H}^{\text{local}}$ into $\mathbf{H}^{\text{global}}$, where each position's representation ($\mathbf{H}_i^{\text{global}}$) incorporates global duplex context. For example, an insertion at g15 attenuates attention weights between distal positions, mimicking the torsional strain observed in PIWI-RNA structures. This architecture mirrors PIWI's dual requirements: localized precision for catalysis and global adaptability for target recognition.

## Insertion feature embedding and prediction head

Insertions can subtly disrupt RNA duplex structure, impacting cleavage efficiency in ways standard pairing features might miss. PAIRNet addresses this by embedding insertion indicators separately, ensuring these effects are fully accounted for:

- **Insertion Embedding:** The insertion indicators are embedded:

$$\mathbf{z} = \mathbf{W}^I \mathbf{I} + \mathbf{b}^I, \quad \mathbf{W}^I \in \mathbb{R}^{27 \times d'}$$

- **Prediction Head:** Final prediction combines all features, where ‖ denotes the feature concatenation:

$$\hat{k} = \mathbf{W}^2 \cdot \text{ReLU}(\mathbf{W}^1 [\text{Flatten}(\mathbf{H}^{\text{global}}) \| \mathbf{z}] + \mathbf{b}^1) + \mathbf{b}^2$$

where $\mathbf{W}^1 \in \mathbb{R}^{(26 \times d + d') \times 34}$, $\mathbf{W}^2 \in \mathbb{R}^{64 \times 1}$.

## Interpretable positional impact analysis

To turn predictions into biological understanding, PAIRNet includes interpretability tools—gradient saliency and counterfactual analysis—that reveal how specific positions and mismatches shape cleavage, fulfilling the promise of actionable insights for piRNA design.

**Gradient saliency mapping.** To identify which guide RNA positions most critically influence cleavage efficiency, PAIRNet employs gradient-based saliency maps. These maps spotlight the input features—specifically, the guide RNA positions—that, when altered, would most significantly affect the predicted cleavage rate, as determined by the magnitude of the gradients. This method harnesses the intuitive power of gradients to reveal how sensitive the model's predictions are to changes at each position, offering a clear window into PAIRNet's decision-making process. We favor gradient-based saliency maps for their computational simplicity and direct interpretability, perfectly suited to unraveling the positional intricacies of RNA interaction modeling. For input features $\{\mathbf{G}, \mathbf{P}, \mathbf{I}\}$, the saliency score $S_i$ for position $i$ is computed as:

$$S_i = \mathbb{E}_{\mathcal{B}}\left[\sum_{d=1}^{4} \left|\frac{\partial \hat{k}}{\partial G_{id}}\right|\right], \quad i \in \{1, \dots, 26\}$$

where $G \in \mathbb{R}^{B \times 26 \times 4}$ represents a batch of $B$ guide RNA sequences, $G_{id}$ denotes the presence (1) or absence (0) of nucleotide $d$ at position $i$, and $\mathbb{E}_{\mathcal{B}}$ averages scores across the batch. Higher $S_i$ values highlight positions (e.g., g10–g11) where minor perturbations drastically reduce cleavage rates, consistent with their role in catalytic metal ion coordination.

**Counterfactual mismatch impact.** To assess the influence of mismatches at each guide RNA position, PAIRNet leverages counterfactual analysis, a causal inference technique that evaluates 'what-if' scenarios. By simulating the effect of replacing a mismatch with a perfect match at a specific position—while holding all else constant—this approach isolates the positional contribution to cleavage rate predictions, making it an intuitive choice for analyzing position-specific importance. For each position $i$, the impact $\Delta_i$ is calculated as:

$$\Delta_i = \frac{1}{|\mathcal{M}_i|} \sum_{s \in \mathcal{M}_i} \left[ f(\mathbf{G}^{(s)}, \mathbf{P}^{(s)}_{\text{match},i}, \mathbf{I}^{(s)}) - f(\mathbf{G}^{(s)}, \mathbf{P}^{(s)}, \mathbf{I}^{(s)}) \right]$$

where $\mathcal{M}_i$ is the subset of samples with mismatches at $i$, and $\mathbf{P}^{(s)}_{\text{match},i}$ adjusts the pairing state at $i$ to a perfect Watson-Crick match. This analysis reveals that $\Delta_i$ peaks at catalytic positions (e.g., g10–g11) and diminishes toward the 3' end, recapitulating PIWI's tolerance for distal mismatches observed in structural studies. Together, these interpretability tools transform PAIRNet from a black-box predictor into a hypothesis-generating framework, enabling researchers to dissect how specific mismatches or structural perturbations influence PIWI's cleavage logic.

## Training objective

We present a composite loss function to measure the cleavage rate prediction error, it integrates two complementary objectives as follows:

$$\mathcal{L} = \alpha \cdot \underbrace{\frac{1}{N} \sum_{i=1}^{N} |k_i - \hat{k}_i|}_{\text{MAE}} + (1 - \alpha) \cdot \underbrace{\left( 1 - \frac{\mathbb{E}[(k_i - \bar{k})(\hat{k}_i - \bar{\hat{k}})]}{\sigma_k \sigma_{\hat{k}}} \right)}_{\text{Pearson Loss}},$$

where: MAE (Mean Absolute Error) quantifies average absolute deviation between experimental rates $k_i$ and predictions $\hat{k}_i$; Pearson Loss $(1 - \text{PCC})$ measures deviation from perfect linear correlation, with $\bar{k}$ and $\sigma_k$ denoting mean and standard deviation of experimental rates. The hyperparameter $\alpha \in [0, 1]$ balances precision (MAE) versus rank consistency (Pearson). We optimized $\alpha$ through grid search, and the optimal value ($\alpha = 0.8$) maximized performance on holdout data, emphasizing MAE minimization (critical for quantitative rate prediction) while retaining sufficient Pearson correlation for comparative guide ranking.

## Datasets

We collect the CNS-seq data in the previously published paper [14] to train the PAIRNet. Four synthetic guide RNAs, namely L1MC [24], Kctd7 [25], piRNA1(piRNA identical to let7-a [24]) and piRNA2 (piRNA targeting luciferase ORF of pGL2 plasmid [24]) were loaded to mouse MIWI and MILI proteins to assemble piRISC. These four guide RNA correspond to dataset1 to dataset4, respectively. Each cleavage assay of corresponding DNA-blocking oligonucleotides target was conducted under specific buffer conditions with GTSF1 and activate piRISC at 33°C for time series. The cleavage reactions were quenched, followed by RNA extraction, reverse transcription, cDNA amplification and sequencing. Data from multiple trials were combined to estimate pre-steady-state cleavage rates. The estimated pre-steady-state cleavage rate $k$ ($k_2 + k_3$), was reported. Both the detailed wet-lab experiments details and data analysis codes could be found in papers [14,15]. The ready-to-use data (target sequence, pairing mode and cleavage rate) can be downloaded from paper's supplementary files [14,15]. Calculated $k$ value was used as label for regression. Additionally, a mixed dataset was constructed by pooling all four guide RNA datasets and splitting them into 75% training and 25% testing sets, complementing the hold-out dataset evaluations.

## Experiment settings

To validate PAIRNet's design, we conducted a two-stage analysis. First, interaction-centric encoding was benchmarked against sequence-based strategies (concatenated one-hot vectors, $K$-mer frequencies, and CNN-based embedding) using conventional models: XGBoost [26], Random Forest [27], MLP [28], Gradient Boosting [29], and FFNN [30]. After confirming its superiority, we compared PAIRNet's hybrid architecture—using the same interaction-centric encoding—to these baselines alongside additional methods (kNN [31], Ridge Regression [32], Bagging [33], AdaBoost [34], Extra Trees [35]).

For the second stage, models were evaluated via Leave-One-Guide-Out cross-validation. In this setup, one guide RNA (and all its associated targets) was completely excluded from the training set and reserved for testing to assess the model's ability to generalize to unseen guide sequences. Cleavage rates ($k$) were standardized using z-score normalization (mean=0, variance=1) during training. The best epoch was selected by minimizing the average validation MAE across folds, with final models retrained on the full training set and tested on held-out data. Training used AdamW (learning rate $10^{-4}$, weight decay 0.1) with a hybrid MAE-Pearson loss ($\alpha = 0.8$ for MAE, $1 - \alpha = 0.2$ for Pearson) and a learning rate scheduler that reduced the rate by a factor of 0.5 if validation loss plateaued for 5 epochs. All experiments ran on one NVIDIA TITAN V GPU with a batch size $B = 128$. Experiments were repeated 10 times with different random seeds to compute mean ± SD for PCC. Statistical significance ($p < 0.05$) was determined via paired $t$-tests between PAIRNet and baselines. PAIRNet processes guide RNAs (26 nucleotides) and targets through position-aware embeddings ($d = 16$, $W^G \in \mathbb{R}^{26 \times 16}$) and insertion embeddings ($d' = 8$, $W^I \in \mathbb{R}^{27 \times 8}$), combining a 1D convolutional layer (16 filters, kernel size 3) with a 3-layer Transformer (4 attention heads). Four ablated variants were tested: NoPosition (removes positional embeddings), PureCNN (discards Transformer), SimplePairing (binary match/mismatch), and NoInsert (ignores insertions).

## Results

To validate PAIRNet's innovations, we structured our analysis into two stages (see Experiment Settings). First, we asked: does interaction-centric encoding outperform sequence-based approaches? Testing five conventional models revealed a critical flaw in traditional methods: their inability to model RNA interaction dynamics. With this established, we deployed PAIRNet's hybrid CNN-Transformer architecture—using the same encoding—to demonstrate superiority over baselines. This isolates PAIRNet's dual strengths: a novel encoding strategy and its ability to integrate local and global RNA interactions.

### Interaction-centric encoding outperforms sequence-based methods

PAIRNet's interaction-centric encoding strategy demonstrated dominant performance across all datasets, models, highlighting its unique suitability for modeling RNA-guided cleavage. This superiority is particularly evident where conventional sequence-based methods (CNN, Concatenation, $K$-mer) exhibited complete predictive failure with zero or negative correlations. The performance comparisons are reported as barplots in Fig 3, with detailed results provided in S1 Table.

For **MIWI** (Fig 3A), interaction-centric encoding demonstrated systematic improvements over conventional approaches. In XGBoost dataset3, interaction-centric strategy achieved PCC 0.688, notably outperforming both CNN ($0.014 \pm 0.0.064$) and Concatenation ($-0.155 \pm 0.000$) with absolute PCC gains of 0.674 and 0.843 respectively. Notably, in dataset1, the interaction-centric FFNN achieved a positive correlation, whereas CNN encoding exhibited a negative correlation, reflecting the consistent advantage of interaction-centric feature engineering in capturing global guide-target dynamics over CNN's localized motif extraction—a pattern corroborated across multiple datasets (e.g., MILI dataset3: FFNN 0.203 vs. CNN $-0.033$). For **MILI** (Fig 3B), the framework exhibited consistent superiority across model architectures. Interaction-centric Random Forest achieved the highest PCC in dataset3, contrasting with CNN's near-zero performance and Concatenation's failure. Notably, $K$-mer encoding showed detrimental effects in dataset1 ($-0.133 \pm 0.011$) and dataset2

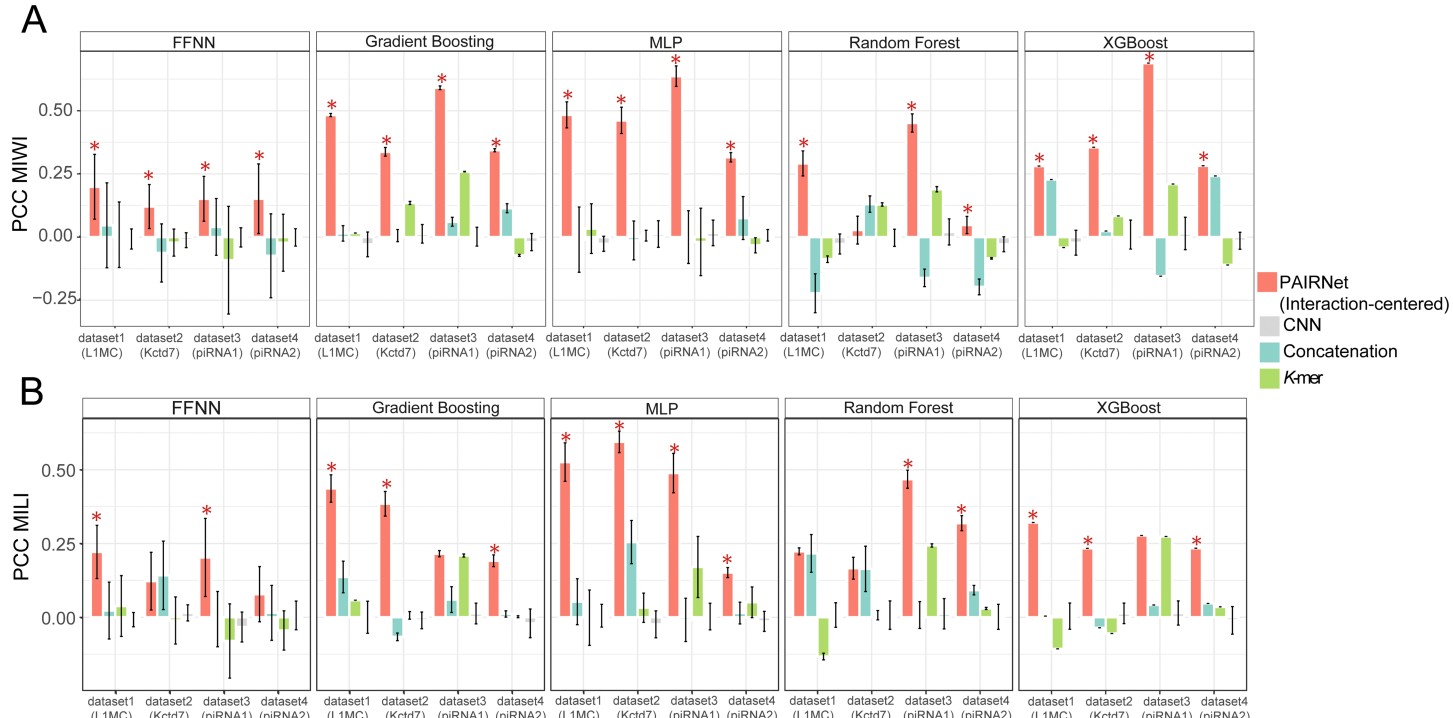

**Fig 3. Interaction-centric encoding outperforms sequence-based methods across PIWI homologs.** Bar plots compare the cleavage rate prediction performance (PCC, mean ± SD, $n = 10$) of interaction-centric encoding versus sequence-based methods (CNN, $K$-mer, concatenation) for five models (MLP, Gradient Boosting, XGBoost, FFNN) across four datasets (1–4) for (A) MIWI and (B) MILI. Red stars (*) indicate interaction-centric models with statistically significant superiority over the second-best method (t-test, $p < 0.05$). Dataset labels correspond to guide RNAs: L1MC (1), Kctd7 (2), piRNA1 (3), piRNA2 (4). $K = 3$ was implemented in $K$-mer encoding. The source data is in S1 Table.

(0.007 ± 0.016), highlighting the risks of sequence-only approaches. The most stable enhancement emerged in MLP dataset2, where interaction-centric encoding surpassed Concatenation by 0.339 PCC with reduced variability.

Across all comparisons (40 model-dataset pairs, 5 models for 4 datasets with MILI/MIWI), interaction-centric encoding significantly outperformed sequence-based methods in 82.5% of cases (33/40). Sequence-based methods exhibited high variability, while interaction-centric predictions remained relatively stable. This universal superiority underscores that explicitly modeling guide-target interactions—rather than treating sequences as independent entities—is essential for predicting PIWI cleavage efficiency.

## PAIRNet achieves robust generalization

Building on the demonstrated superiority of interaction-centric encoding, PAIRNet integrates this strategy with a hybrid CNN-Transformer architecture to further resolve local and global determinants of cleavage efficiency. PAIRNet demonstrates robust generalization capabilities across both hold-out and mixed datasets, achieving state-of-the-art predictive accuracy with the most pronounced improvements of 34.7% for MILI (dataset3) and 14.6% for MIWI (dataset2) over second-ranking methods in PCC (Fig 4 and S2 Table). These substantial margins highlight its unique capacity to resolve complex interaction patterns, such as catalytic core complementarity and structural perturbations, while maintaining consistent superiority in simpler contexts.

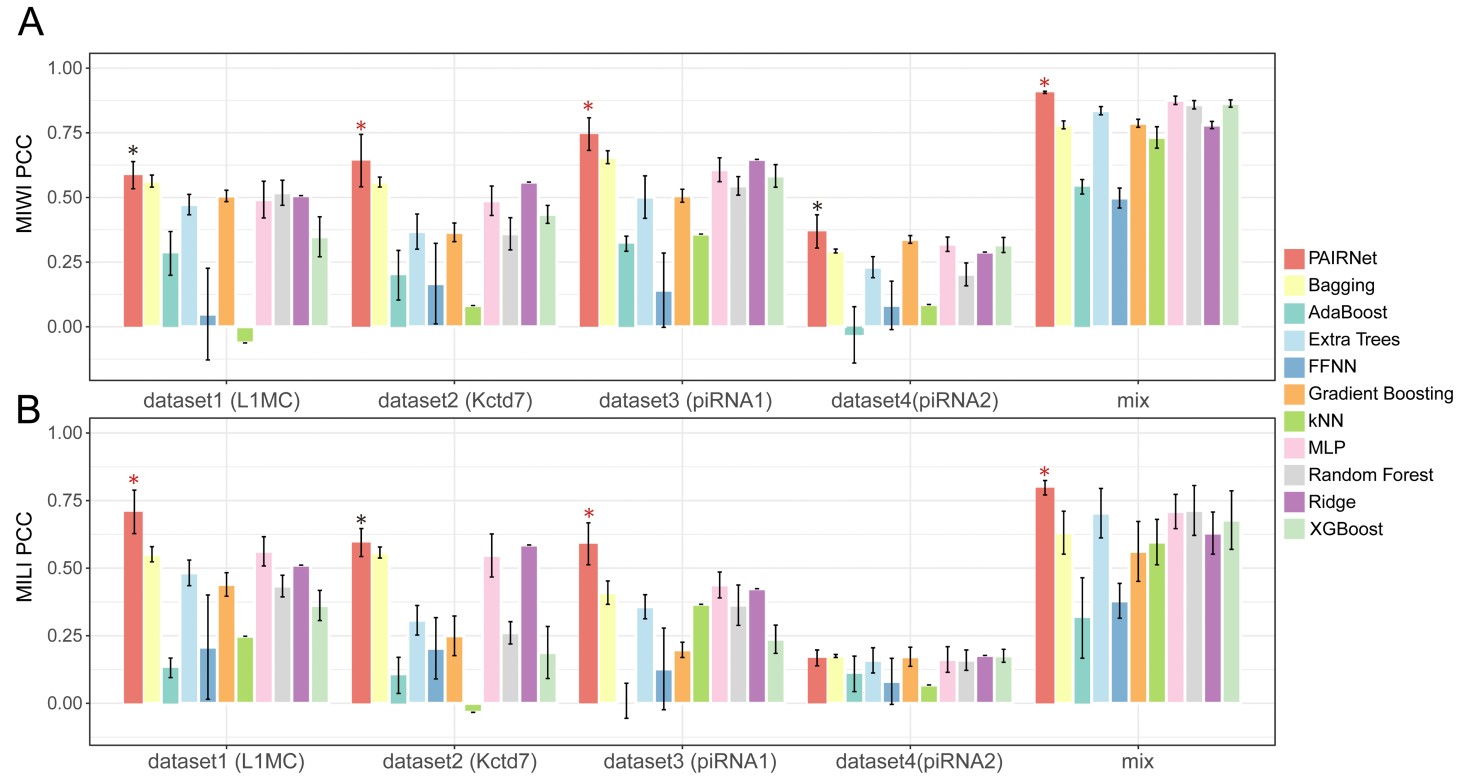

**Fig 4**. **PAIRNet outperforms conventional methods across PIWI homologs and datasets.** Bar plots compares PAIRNet with ten baseline methods on hold-out datasets (1–4) and mixed data for (A) MILI and (B) MIWI (PCC, mean $\pm$ SD, $n = 10$). Error bars represent variability across 10 independent replicates. Stars denote statistical significance (paired $t$-test): black (highest performer, $p \geq 0.05$), red ($p < 0.05$). Datasets correspond to guide RNAs: L1MC (1), Kctd7 (2), piRNA1 (3), piRNA2 (4). Mixed dataset indicates the testing split from pooled guide RNAs. The source data is in S2 Table.

For MIWI (Fig 4A), dataset1 showcases the most pronounced gain: PAIRNet (0.586±0.052) outperforms kNN (−0.062± 0.000) by a PCC margin of 0.648—the highest absolute improvement observed. In dataset3, PAIRNet achieves the highest PCC (0.745±0.063) across all MIWI datasets, exceeding MLP by 0.138. For MILI (Fig 4B), dataset3 exhibits the widest margin: PAIRNet surpasses XGBoost by 0.353 PCC. Even in dataset4—the most challenging MILI dataset with universally low PCC values—PAIRNet (0.168 $\pm$ 0.029) retains lower variability than Gradient Boosting, underscoring its reliability under adverse conditions.

Building on these results, PAIRNet demonstrates robust generalization across four diverse hold-out datasets, achieving superior predictive accuracy for MILI and MIWI with mean PCC ranges of 0.168–0.797 (MILI) and 0.369–0.906 (MIWI). These ranges not only exceed baseline models—particularly tree-based methods like Random Forest (0.160–0.714) and XGBoost (0.176–0.863), which exhibit lower and less stable performance—but also highlight PAIRNet's adaptability to varied interaction complexities. Statistical validation further solidifies this advantage, with $p < 0.05$ significance in 80.0% (40/50) of MILI and 96.0% (48/50) of MIWI comparisons. While simpler models like kNN occasionally yield negative correlations (–0.12–0.31), PAIRNet maintains reliable performance even in non-significant edge cases (e.g., against Bagging in one MILI dataset), underscoring its capacity to decode PIWI's dynamic RNA interactions across experimental contexts.

### Ablation studies validate PAIRNet's core components

To dissect the contributions of PAIRNet's architecture, we systematically ablated key components and assessed their impact on cleavage rate prediction across mixed and hold-out datasets for MIWI and MILI (Fig 5 and S3 Table). We tested

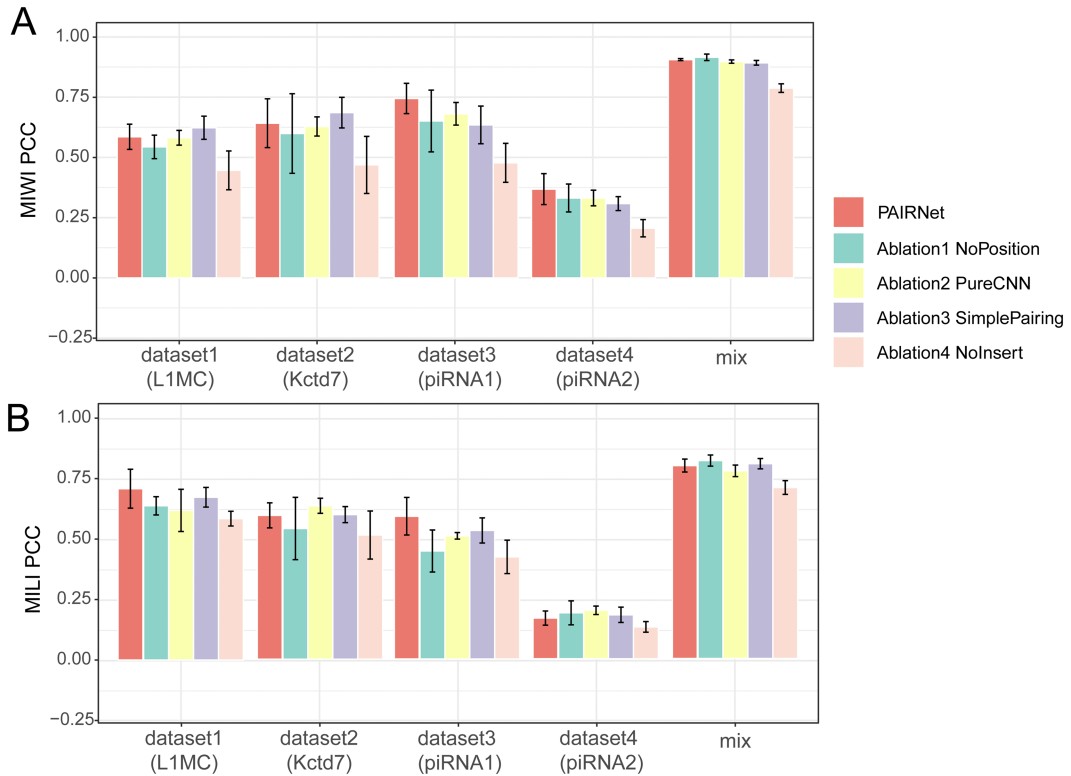

**Fig 5. Ablation Study of PAIRNet Across Datasets.** Bar plots illustrating the mean PCC for the full PAIRNet model and its ablation variants (NoPosition, PureCNN, SimplePairing and NoInsert) on MIWI (A) and MILI (B) datasets, across four hold-out datasets (dataset1–dataset4) and a mixed dataset (mix) (PCC, mean $\pm$ SD, $n = 10$). Error bars represent standard deviations, highlighting the impact of each component on model performance. NoPosition removes positional embeddings, stripping spatial context; PureCNN discards Transformer, losing long-range dependency modeling; SimplePairing simplifies pairing features to a binary match or mismatch, reducing mismatch detail; and NoInsert omits insertion handling, ignoring structural disruptions. Mixed dataset indicates the testing split from pooled guide RNAs. The source data is in S3 Table.

four variants: NoPosition removes positional embeddings, stripping spatial context; PureCNN removes the Transformer layers and directly flattens the output after the convolutional layer, losing long-range dependency modeling; SimplePairing simplifies pairing features to a binary match or mismatch, reducing mismatch detail; and NoInsert omits insertion handling, ignoring structural disruptions. These ablations reveal how each element drives PAIRNet's performance.

Ablation studies underscore the necessity of PAIRNet's design. Removing positional embeddings (NoPosition) reduced predictive accuracy in critical hold-out datasets (e.g., MIWI dataset3: PAIRNet $0.745 \pm 0.063$ versus NoPosition $0.652 \pm 0.128$, Fig 5A), while removal of Transformer (PureCNN) impaired global interaction modeling (MILI dataset3: PAIRNet $0.590 \pm 0.078$ versus PureCNN $0.509 \pm 0.014$, Fig 5B). Simplifying mismatch encoding (SimplePairing) or omitting insertions (NoInsert) further degraded performance, emphasizing the importance of biochemical specificity. NoPosition achieved marginally higher PCC on mixed datasets (MIWI: $0.916 \pm 0.014$ versus PAIRNet $0.906 \pm 0.004$, Fig 5A). However, saliency maps revealed a critical distinction: NoPosition distributed importance uniformly across all guide positions (Fig 6B), failing to prioritize catalytic residues (g9–g11) essential for PIWI cleavage. In contrast, PAIRNet retained robust predictive stability (e.g., MIWI mix: SD = 0.004 versus NoPosition SD = 0.014) while aligning saliency scores with structural insights—a dual strength absent in ablated variants (Fig 6A). Thus, PAIRNet's consistent superiority on hold-out datasets, coupled with its enhanced explainability, underscores its effectiveness for unseen guide RNAs.

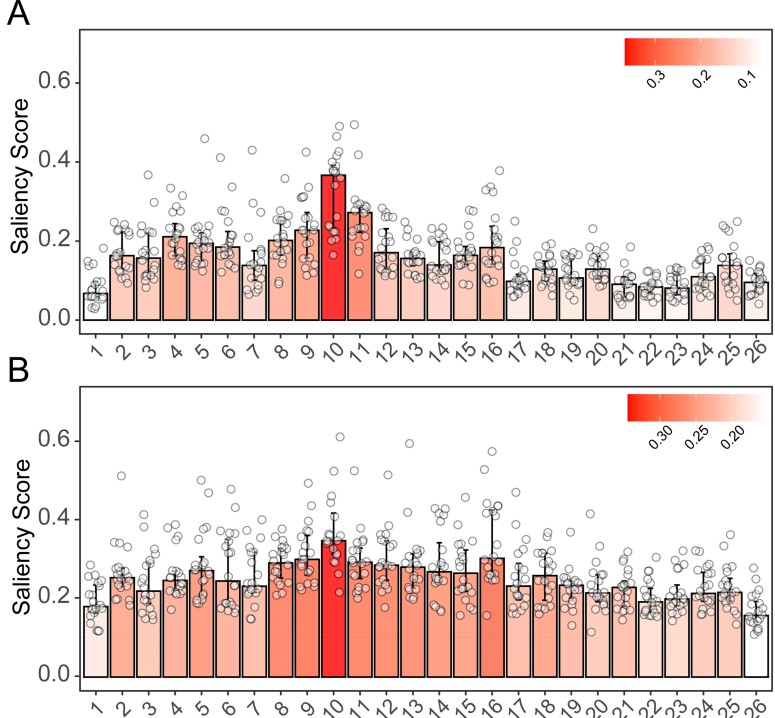

**Fig 6. Feature importance comparison between the full PAIRNet model and the NoPosition ablation variant.** Bar plots show the saliency score of each position of guide RNA using mixed model, integrating data from 4 guides of MILI and MIWI (Saliency score, mean ± SD, $n = 10$). (A) With Position Embedding: Saliency scores derived from the full PAIRNet model, which includes learnable positional encodings. (B) NoPosition (Ablation): Saliency scores derived from the ablated model where position embeddings were removed, showing a uniform distribution of importance due to the loss of spatial context.

In short, while ablated variants may occasionally excel in specific contexts, the full PAIRNet model delivers robust, generalizable performance across diverse datasets, cementing its value for PIWI cleavage prediction.

## Position-specific cleavage rules align with structural and catalytic insights

We used counterfactual analysis to study the impact on the cleavage rate imposed by the mismatch. The counterfactual inference on position mismatch was then carefully aligned to the structural property and functional domain, stressing the interpretability of our model (Fig 7). The model first correctly de-emphasizes the 5'-most position of guide RNA (Fig 7 blue), whose physical position within piRISC was well-defined as anchored to a specialized pocket in the middle (MID) domain of Argonaute proteins [23]; next the result showed intermediate importance for g2 to g8 (required canonical uninterrupted seed pairing for animal AGOs), which is dispensable for PIWI cleavage [14], and from structure perspective showing less critical for target recognition due to PIWI's relaxed pairing rules, related research explained that PIWI-clade Argonautes feature a wider channel that accommodates more extensive base pairing beyond the seed region, which enhances targeting accuracy while allowing some degree of target mispairing [16,22,23,36]; our model highlights the most important positions as g9, g10 and g11 (Fig 7 yellow), which is indeed the PIWI protein's catalytic core, which is indispensable for cleavage activity, these positions form the 'slicer' site and are stabilized by extensive interactions with the PIWI domain's U-loop, mutating residues that interact with the guide RNA backbone between g8 and g11 (that is, L1 residues S379A, H380A and K381A, or PAZ-domain residues T424A and T499A), or residues that interact with central positions of the duplex (that is, L1 hairpin R339G and N340G or cS4 loop S816A, D817A, and Q819A) all reduced target

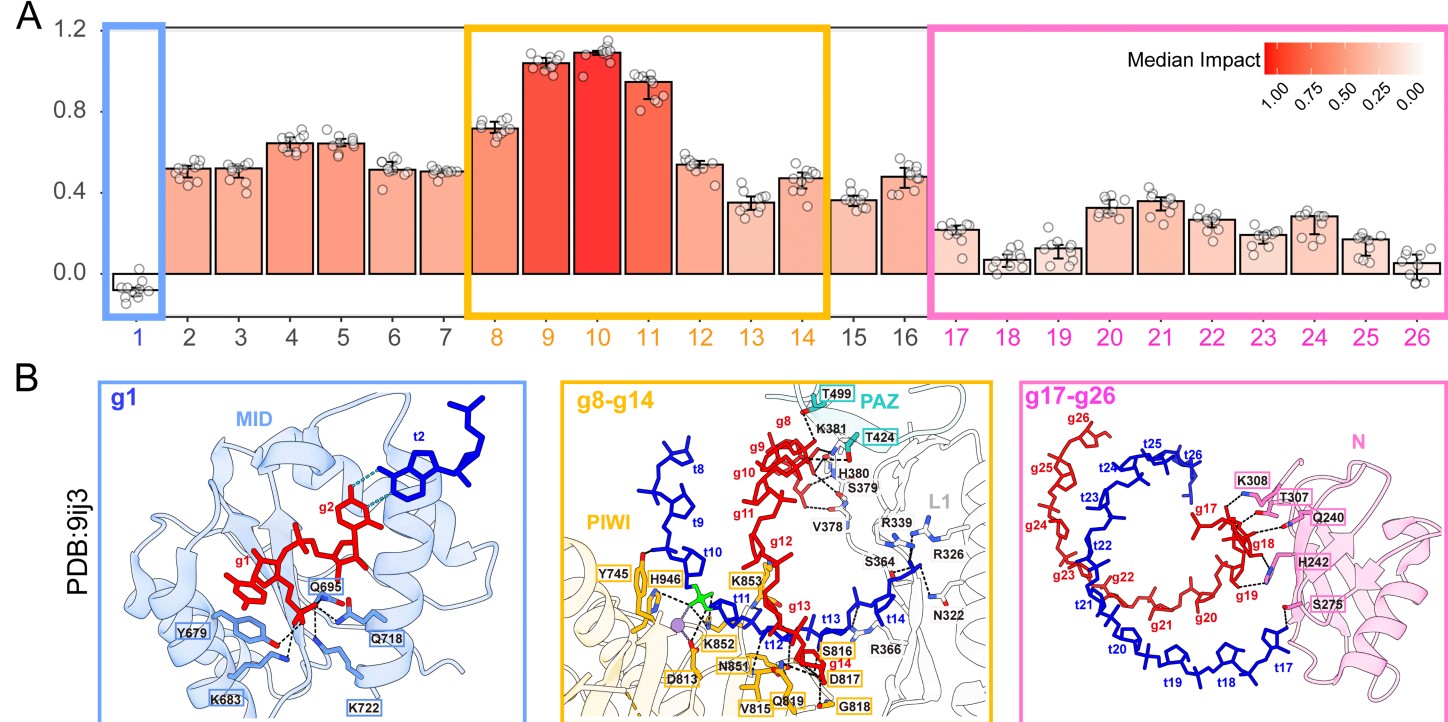

**Fig 7**. **Alignment of Position Impact by Counterfactual Analysis Result to Structural Insight of piRISC.** (A) showed median impact score of each 1-based guide position from g1 to g26 derived from the MILI-specific PAIRNet model; (B) showed three sections (g1, g8 to g14 and 3'end) were captured from Cryo-EM-derived 3D structure with PDB entry 9ij3 [22] (See S2 Fig for Counterfactual Analysis integrating MILI and MIWI).

cleavage activity of MILI [22], highlighting the importance of these contacts in MILI-mediated target cleavage [22], lastly, there's downshift from 16th position, which is 3'end of guide RNA (Fig 7 pink), our model identified reduced importance of this region, and related work showed pairing to piRNA 3' end is dispensable for PIWI slicing, and from structural perspective, the 3' end of the guide RNA (positions 20–26) is exposed in the locked state. While necessary for initial binding, mismatches here have minimal impact on cleavage efficiency, The PAZ domain initially binds the guide RNA's 3' end but releases it upon duplex extension, reducing reliance on seed pairing [22]. Actually, structural research captured the structures of piRISCs bound to target RNAs of increasing length and perfect complementarity to guide strand nucleotides g2–g8, g2–g15 or g2–g26. Our model showed similar 3-stage importance for this pairing extension.

The PAIRnet model's positional importance profile (Fig 7A) accurately recapitulates the dynamic conformational changes of PIWI proteins during target cleavage (Fig 7B). By emphasizing positions critical for duplex stabilization (central region) and catalytic activation (positions 10–11), while downplaying non-essential regions (seed and 3' end), the model aligns with the biological mechanism described in the paper. This validates its utility for interpreting piRNA targeting rules and transposon silencing dynamics.

## Position-specific cleavage rules align with structural and catalytic insights

PAIRNet generalizes to AGO2 and captures distinct lineage-specific targeting rules. To evaluate PAIRNet's ability to generalize beyond PIWI proteins, we retrained the framework on mouse AGO2 Cleave-N'-Seq data. Unlike PIWI proteins, which utilize a 'relaxed' targeting mechanism, AGO2 enforces stricter geometric constraints, particularly in the central cleft [14,15]. The AGO2-specific model achieved a PCC of $0.769 \pm 0.028$. Critically, comparative interpretability analysis

revealed a sharp divergence in feature importance: while the PIWI model displays distributed importance compatible with mismatch tolerance, the AGO2 model exhibits a high-magnitude saliency peak centered on the catalytic core and central region (g10-g15) in S3 Fig. This aligns with structural evidence that AGO2 is highly sensitive to mismatches near the scissile phosphate, confirming that PAIRNet captures the distinct biophysical fingerprints of different Argonaute families rather than memorizing a single rule set [14,15].

## Conclusion

PIWI proteins safeguard genome integrity through piRNA-guided RNA cleavage, a process governed by dynamic guide-target interactions. While CNS-seq has advanced our understanding of PIWI targeting logic, its experimental limitations hinder systematic exploration of sequence determinants. PAIRNet integrates interaction-centric encoding to define what pairing geometries matter, positional embeddings to pinpoint where catalytic residues dominate, and a hybrid CNN-Transformer architecture to resolve how local and global interactions collectively govern cleavage efficiency—establishing a comprehensive framework that outperformed conventional sequence-based approaches by up to 34.7% (MILI) and 14.6% (MIWI) in Pearson correlation.

Complementing this encoding, PAIRNet's hybrid architecture resolves both local catalytic motifs (e.g., contiguous pairing at g10–g11) and global structural perturbations (e.g., distal insertions), achieving state-of-the-art accuracy with pronounced improvements in key datasets. Interpretability modules further bridge computational predictions with biological reality: saliency maps prioritize catalytic residues validated by structural studies, while counterfactual analysis quantifies mismatch tolerance at non-essential regions.

By transcending CNS-seq's throughput limits, PAIRNet accelerates the design of high-specificity piRNA guides for transposon silencing and antiviral defense. Future work could extend this framework to model *in vivo* contexts, integrate 3D structural data for refined positional dependencies, or adapt its interaction-centric principles to other RNA-guided silencing systems (e.g., CRISPR-Cas). PAIRNet's success demonstrates that biochemical precision, not just sequence, dictates RNA function—a paradigm shift with broad implications for genome engineering and programmable RNA therapeutics.

## Discussion

PAIRNet's hybrid CNN-Transformer architecture and interaction-centric encoding were deliberately crafted to mirror PIWI's dual requirements: local catalytic precision (e.g., stringent complementarity at g10–g11) and global structural adaptability (e.g., tolerance for 3' mismatches). By explicitly modeling Watson-Crick pairing, mismatch identities, and indels—augmented with learnable positional embeddings—PAIRNet quantifies how spatial hierarchies in RNA duplexes dictate cleavage efficiency. The CNN layers act as molecular sensors, detecting contiguous base-pairing motifs essential for catalytic activation, while transformer layers resolve long-range structural perturbations (e.g., bulges from insertions), mirroring PIWI's conformational dynamics. This design aligns with structural studies showing that PIWI-clade Argonautes enforce rigid pairing at catalytic residues while accommodating flexibility elsewhere, enabling both specificity and adaptability in transposon silencing.

Traditional machine learning approaches (e.g., Random Forest, XGBoost) falter because they treat RNA sequences as position-independent feature sets, overlooking the spatial and structural dependencies central to PIWI targeting. For instance, while these models might flag mismatches, they cannot distinguish between a destabilizing A-A clash at g10 and a benign G-U wobble at g25. Similarly, CNNs prioritize local sequence motifs but fail to contextualize distal structural disruptions (e.g., a g15 bulge reducing cleavage efficiency via allosteric effects). PAIRNet's success lies in its explicit modeling of interaction geometry and positional hierarchy, bridging a critical gap between sequence abstraction and biochemical reality.

PAIRNet's strengths lie in its biological interpretability, high accuracy, and scalability. By quantifying position-specific mismatch penalties and insertion impacts, it provides actionable rules for designing piRNA guides with enhanced specificity. However, there are still limitations. The model is trained on in vitro CNS-seq data, which may not fully recapitulate in vivo complexity (e.g., chromatin effects, co-factors like GTSF1). Furthermore, PAIRNet is trained on mammalian PIWI data, which reflects a 'relaxed' targeting mechanism. Consequently, the current model may not fully generalize to other systems, such as the silkworm Siwi-RE, where a 'locked' conformation requires extensive 3' pairing to structurally activate the complex [37]. Additionally, the current dataset's limited guide sequence diversity restricts the model's ability to fully generalize intrinsic guide efficacy rules. Future work should expand to in vivo cleavage assays, additional PIWI homologs (e.g., HIWI, HILI), and other species-specific data. Integrating 3D structural data (e.g., PIWI-RNA cocrystal structures) may further refine feature representations. Finally, extending PAIRNet to predict off-target effects or synergize with CRISPR-based systems may unlock new avenues for RNA-guided genome engineering. While PAIRNet effectively flags the structural disruption caused by insertions and deletions, our current encoding does not distinguish the specific nucleotide identity of the inserted base. Future iterations of the framework could incorporate the thermodynamic contributions of specific indel sequences (e.g., differentiating stabilizing vs. destabilizing bulges) to further refine prediction accuracy.

In summary, PAIRNet bridges computational innovation with mechanistic insights into PIWI biology, offering a versatile framework to decode and engineer RNA-mediated silencing pathways.

## Supporting information

**S1 Fig. CNS-seq cleavage analysis for single and double mismatches across PIWI proteins.** Pre-steady-state cleavage rates for targets containing (A) single or (B) two consecutive mismatches between guide positions g2 and g20. Data includes MILI, MIWI, and E$f$Piwi with contiguous pairing extending to g21 or g25. Expanded $k$ distributions for single-mismatch targets across four distinct guide RNAs loaded into (C) MILI and (D) MIWI. Each dot represents a specific mismatch geometry within the g2–g20 region (g2–g21 pairing). Median and interquartile range (IQR) are shown for all panels. (PDF)

**S2 Fig. Counterfactual analysis of position importance using the Mixed Model.** Median impact scores derived from the PAIRNet Mixed Model (integrating both MIWI and MILI datasets). The mixed model retains the feature importance profile centered on the catalytic core (g10–g11), though with higher variance due to subtle kinetic differences between PIWI homologs. (PDF)

**S3 Fig. Position importance analyzed for AGO2 and MILI/MIWI by saliency scoring.** Saliency score are calculated for MILI/MIWI (A) and AGO2 (B), Position 3 and 13 are important positions mentioned by Becker et al. [15]. (PDF)

**S1 Table. Source data for encoding strategies performance comparisons.** (XLSX)

**S2 Table. Source data for prediction performance comparisons.** (XLSX)

**S3 Table. Source data for ablation study.** (XLSX)

## Author contributions

**Conceptualization:** Enzhi Shen, Lei Xu.

**Data curation:** Lin Zeng, Zhenzhen Li.

**Formal analysis:** Lin Zeng, Zhenzhen Li.

**Funding acquisition:** Shikui Tu, Lei Xu.

**Investigation:** Lin Zeng, Zhenzhen Li.

**Methodology:** Lin Zeng.

**Supervision:** Enzhi Shen, Shikui Tu.

**Validation:** Lin Zeng.

**Visualization:** Lin Zeng, Zhenzhen Li.

**Writing – original draft:** Lin Zeng.

**Writing – review & editing:** Lin Zeng, Enzhi Shen, Shikui Tu.

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
