## [Decision Letter · Decision Letter 0]

7 Nov 2025

PCOMPBIOL-D-25-00988

PAIRNet: Predicting PIWI cleavage specificity via position-aware RNA interaction modeling

PLOS Computational Biology

Dear Dr. Tu,

Thank you for submitting your manuscript to PLOS Computational Biology. After careful consideration, we feel that it has merit but does not fully meet PLOS Computational Biology's publication criteria as it currently stands. Therefore, we invite you to submit a revised version of the manuscript that addresses the points raised during the review process.

We look forward to receiving your revised manuscript.

Kind regards,

Nir Ben-Tal

Section Editor

PLOS Computational Biology

**Journal Requirements:**

At this stage, the following Authors/Authors require contributions: Lin Zeng, Zhenzhen Li, Enzhi Shen, Shikui Tu, and Lei Xu. Please ensure that the full contributions of each author are acknowledged in the "Add/Edit/Remove Authors" section of our submission form.

**Reviewers' comments:**

Reviewer's Responses to Questions

**Comments to the Authors:**

Reviewer #1: The manuscript by Zeng et al describes a new deep learning framework that captures targeting rules by PIWI proteins. This is an important unaddressed question and the authors were able to create a model that mostly recapitulates the biochemical experiments. However, several key questions remain unanswered.

1) Can authors' model explain why there is difference between different guide piRNAs?

2) Cleave-n-Seq data also exists for Ago2, and authors could use their approach to create a second model for AGO proteins and compare the two.

3) Ideally, authors model should be accessible not only as a raw code, but also a convenient tool for prediction of targeting rules for a specific piRNA sequence.

Reviewer #2: This is a study by Shikui Tu and Enzhi Shen’s groups. They build a deep-learning model to investigate the piRNA cleavage rule, trained on Cleave-N’-Seq data. This is the first study using deep learning models to investigate piRNA cleavage rule as I know. The model has two main innovations: 1. It encodes interaction information instead of typical one-hot sequence encode. This enables the corporation of mismatches and indels. 2. The model uses a hybrid CNN-transformer architecture, which captures both the local motifs (consecutive pairing) and the global effect of mismatch/indel. The model overall achieves good performance. The authors also interpreted the model and the results are consistent with known knowledges.

Overall, this work is solid and important for the small RNA silencing field. The manuscript is well-written. However, I have several suggestions concerning the feature encoding below:

Major comments:

1. In Fig 1D. Mismatches are encoded as “A|T|C|G”, how does it performs comparing to simply encode mismatches as “mismatches”? In my opinion, different base pairs has different physical affinity (e.g. C-A are more stable than T-T), encoding mismatches into base pairs would be a better way. The same for insertions and deletions.

2. The last paragraph of Introduction seems to be heavily AI edited, and the impact of PAIRNet is exaggerated.

3. Fig 3 and 4. I am wondering how PAIRNet compares to other encoding methods + the CNN-transformer hybrid architecture. It is no wonder typical tree-based models (random forest etc.) cannot decode sequences into interactions, which is why they all perform bad in Fig 3.

Minor comments:

1. Fig 1A. Mismatch and indel are not present intuitively. I would suggest a dark dot as mismatch, a “–” indicates deletions, and a “-AT-” as insertions.

2. line 61. “When revisiting MILI/MIWI data in paper” should be “When revisiting MILI/MIWI data in Gainetdinov et al.”.

3. line 62-64. It would be better to put Fig S1–4 as different panels, as they support the same conclusion.

4. Fig S1 is clipped.

5. line 64. “we still observed variations 63 between different guide RNA both for MILI (S3 Fig) and MIWI (S4 Fig).” Do you mean Fig S1–2?

6. line66. Cite Figures.

7. Fig. 1C. What is the difference between concatenation- and CNN-based embeddings?

8. line95. Typo: “initeraction”.

9. Fig 6. “NoPosition” and “Position Embedding” should be clearly indicated in the figure title or legend.

Reviewer #3: The manuscript describes a new ML approach to predict rates for piRNA-guided cleavage. Small RNA sequence influences AGO/PIWI-catalyzed cleavage rates (PMID: 39025072 and PMID: 37344600). The important goal of this work appears to be finding a unifying rule set for piRNAs of different sequences. It is a useful study, but several major concerns need to be addressed:

1. If authors hold out one piRNA (e.g., L1MC) and train the model without it, will the 3-piRNA (piRNA#1, piRNA#2, Kctd7) model predict how the fourth piRNA cleaves? The hold-out experiment should be repeated for each of the four piRNAs.

2. Do figures 6 and 7 show data from all models (four datasets and two proteins as well as mixed)? If so, it would be important to highlight the data from the mixed model, as such a model would reflect more general rules

3. Authors could test how their model predicts data from another cleave'n'seq experiment described in PMID: 40912244

**Have the authors made all data and (if applicable) computational code underlying the findings in their manuscript fully available?**

Reviewer #1: **No:** https://github.com/CMACH508/PAIRNet is not reachable

Reviewer #2: Yes

Reviewer #3: Yes

PLOS authors have the option to publish the peer review history of their article (what does this mean?). If published, this will include your full peer review and any attached files.

Reviewer #1: No

Reviewer #2: **Yes:** Tianxiong Yu

Reviewer #3: No

**Figure resubmission:**
---

## [Decision Letter · Decision Letter 1]

22 Jan 2026

Dear Prof Tu,

We are pleased to inform you that your manuscript 'PAIRNet: Predicting PIWI cleavage specificity via position-aware RNA interaction modeling' has been provisionally accepted for publication in PLOS Computational Biology.

Best regards,

Nir Ben-Tal

Section Editor

PLOS Computational Biology

Reviewer's Responses to Questions

**Comments to the Authors:**

Reviewer #2: The authors has addressed all my comments.

Reviewer #3: The authors put great effort to address all comments from all three reviewers and the manuscript has improved significantly. I strongly recommend accepting it and accompanying its publication with the peer review comments and responses as well.

**Have the authors made all data and (if applicable) computational code underlying the findings in their manuscript fully available?**

Reviewer #2: Yes

Reviewer #3: Yes

PLOS authors have the option to publish the peer review history of their article (what does this mean?). If published, this will include your full peer review and any attached files.

Reviewer #2: No

Reviewer #3: **Yes:** Ildar Gainetdinov

---

## [Editor Report · Acceptance letter]

PCOMPBIOL-D-25-00988R1

PAIRNet: Predicting PIWI cleavage specificity via position-aware RNA interaction modeling

Dear Dr Tu,

I am pleased to inform you that your manuscript has been formally accepted for publication in PLOS Computational Biology. Your manuscript is now with our production department and you will be notified of the publication date in due course.

With kind regards,

Anita Estes
